# FC-4DFS: Frequency-controlled Flexible 4D Facial Expression Synthesizing

## ABSTRACT

4D facial expression synthesizing is a critical problem in the fields of computer vision and graphics. Current methods lack flexibility and smoothness when simulating the inter-frame motion of expression sequences. In this paper, we propose a frequency-controlled 4D facial expression synthesizing method, FC-4DFS. Specifically, we introduce a frequency-controlled LSTM network to generate 4D facial expression sequences frame by frame from a given neutral landmark with a given length. Meanwhile, we propose a temporal coherence loss to enhance the perception of temporal sequence motion and improve the accuracy of relative displacements. Furthermore, we designed a Multi-level Identity-Aware Displacement Network based on a cross-attention mechanism to reconstruct the 4D facial expression sequences from landmark sequences. Finally, our FC-4DFS achieves flexible and SOTA generation results of 4D facial expression sequences with different lengths on CoMA and Florence4D datasets. The code will be available on GitHub.

## CCS CONCEPTS

• **Computing methodologies** → **Procedural animation**; Computer graphics; Machine learning.

## KEYWORDS

4D face, neutral landmark, expression generation, LSTM, positional encoding

## 1 INTRODUCTION

4D facial expression synthesizing is a critical problem in the fields of computer vision and graphics and has broad applications in areas such as 3D animations, virtual reality, and interactive gaming. The task aims to generate a series of realistic facial mesh with diverse expressions or speech-related movements, starting from a mesh with a neutral expression.

Although recent advances in generative networks have propelled the development of 2D video animation solutions[12, 25, 34], these videos lack spatial depth, realism, and interactivity, rendering them unsuitable for real-world applications like VR. Afterward, an increasing number of researchers are beginning to explore the driving and generation of 4D facial expression sequences with the introduction of 4D facial expression sequence datasets[8, 9, 29, 30]. Among

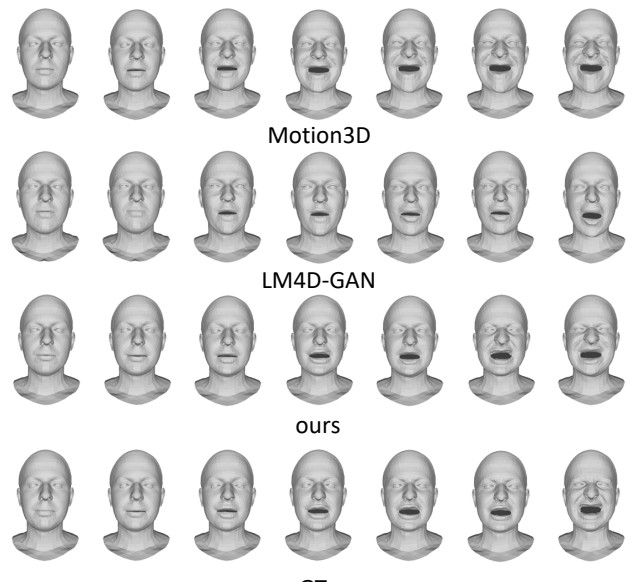

**Figure 1: Quantity generation comparison results between Groud-truth, Motion3D, LM-4DGAN and our FC-LSTM with different identities.**

them, some previous works have focused on generating 4D facial sequences from neutral meshes driven by complex video[5] or audio signals[9, 19, 24]. However, these methods demand substantial prior knowledge and require extensive data, lacking temporal continuity, making direct application in scenarios like game scene development challenging where such priors are absent. In contrast, our research focuses on utilizing expression labels to guide a temporally coherent animation process, enhancing the applicability of facial expression sequence generation in scenarios with minimal priors.

Recently, some researchers introduced GAN[17] into the label-guided 4D facial expression generation task. First, Motion3D[27] proposed the WGAN to generate the Square Root Velocity Function (SRVF) of the 3D landmarks sequences and then they devised an S2D decoder to convert them into mesh displacements, which added to the neutral mesh and yielded a 4D facial expression. This pioneering work has laid the groundwork for its application in a variety of downstream tasks[26]. However, their generated sequences lacked robustness to different identities since they only use expression labels as guides. To address this issue, some recent methods[21, 36, 37] utilize the neutral landmark as inputs to directly generate landmark sequences, thereby enhancing the generation framework's robustness to various identities. However, the sequences generated by these methods exhibit issues such as

sequence motion lacking smoothness (Figure.1 line 1), and missing expression details(Figure.1 line 2). Additionally, their application in real-world scenarios is limited since they can only generate sequences of fixed lengths.

On the other hand, it is essential to transform them into a sequence of 3D meshes after obtaining the sequence of landmarks. N3DMM[3] first introduced a graph convolution method called spiral convolution, to encode and decode entire meshes. However, this method cannot effectively model the neutral mesh of unknown identities. Furthermore, S2D[27] proposed decoding mesh displacements from landmark displacements to reduce the modeling error for meshes of unknown identities. They divide the expression mesh into neutral mesh and displacement components and eliminate the need to encode and decode the dominant neutral mesh body. However, due to the limited information contained in landmark displacements and the diverse identity information, their ability to generalize the expression mesh details across different identities still requires further enhancement.

In this paper, we propose a frequency-controlled 4D facial expression synthesizing method, FC-4DFS, based on frequency-controlled LSTM and multi-level identity-aware network. Specifically, we introduce a frequency-controlled LSTM (FC-LSTM) to generate 4D facial expression landmark sequences frame by frame from a given neutral landmark. Among them, we first modified the basic structure of LSTM to produce different length sequences with controlled frequency. Meanwhile, we integrate the positional information to enhance the awareness of the current frame's position within the sequence. Furthermore, we propose a temporal coherence loss to enhance the perception of temporal sequence motion and improve the accuracy of relative displacements.

On the other hand, we design a multi-level identity-aware network(MIADNet) based on a cross-attention mechanism to reconstruct the 4D facial expression sequences from landmark sequences. Our MIADNet can significantly improve the robustness of various identities and facilitate the generation of detailed and identity-consistent facial expressions. Finally, our FC-4DFS achieves flexible and SOTA generation results of 4D facial expression sequences with different lengths on both the CoMA and Florence4D datasets. In summary, we make the following contributions:

- We introduce a frame-by-frame 4D facial expression generation framework based on the FC-LSTM network and add temporal loss to achieve flexible control of the length of the generated sequence and enhance the smoothness of sequence motion.
- We design the MIADNet, which leverages the cross-attention mechanism and fully utilizes the multi-level identity information of the neutral mesh and neutral landmark, further enhancing the decoding robustness across various identities.
- Our FC-4DFS enables the generation of 4D face expression sequences with different lengths while maintaining sequence integrity and facial details. We also achieve SOTA generation results on both the CoMA and Florence4D datasets.

## 2 RELATED WORK

Here, we review some of the latest advancements in related domains, including 4D facial expression sequence generation and 3D face modeling.

### 2.1 4D facial expression generation

Many researchers are committed to directly modeling 4D facial expression information[6, 7, 16]. However, because mesh has dense vertex information, it is difficult to directly drive mesh through tags to form expression sequences, and cannot effectively simulate expression details[28]. Thanks to the development of neural networks, the detection of facial landmarks is reliable and accurate [11, 13, 32, 35], while landmarks and their motion provide a feasible way to explain facial deformation by simplifying the complexity of visual data. Researchers began to study the use of landmarks to guide expression sequence generation. The work of landmark-guided 2D expression generation has been widely studied[10, 31, 33, 34]. These methods have demonstrated the potential of using landmarks to simulate expression dynamics and generate 2D videos using generative models[17, 18, 20], so researchers began to focus on landmark-guided Generation of 4D expression sequences. Motion3D[27] takes a step further based on MotionGAN[25] and proposes WGAN to generate the square root velocity function (SRVF) of 3D landmark sequences, and then they design an S2D decoder to convert them into mesh displacements, This displacement is added to the neutral mesh and generates a 4D expression, which is applied in downstream tasks[26]. However, the sequences they generated lacked robustness to different identities when generalizing. In order to enhance the generalization ability of the generated sequence to different identities, [21, 36, 37] uses neutral landmark sequences as input to generate complete landmark expression sequences. However, these models can only generate one[36, 37] or several [21] fixed-length sequences and cannot be effectively applied in actual game modeling scenarios.

### 2.2 3D Face Modeling

In order to obtain mesh information from the generated landmark sequences, researchers use the 3D Morphable Model (3DMM) as a bridge between landmarks and 3D facial expressions. 3DMM was originally introduced in [1] and is widely considered the premier framework for 3D facial modeling. The original model and its variants[2, 4, 15, 22, 23] rely on linear formulations to capture changes in facial shape, including identity and expression, thus limiting their modeling capabilities. Ranjan et al[30] introduced an autoencoder architecture based on a newly defined spectral convolution operator, along with pooling operations to facilitate grid down/upsampling. Brisas et al. [3] improve this approach by introducing a novel graph convolution operator that enforces consistent local ordering on the vertices of the graph via a spiral operator. Although their modeling accuracy is impressive across different expressions, recent research [14] highlights their significant shortcomings in generalizing to unknown identities. This severely limits their practical applicability in tasks such as face fitting or expression transfer. Next, S2D[27] begins to decouple neutral meshes and displacements, using landmark displacements to guide the modeling of 3D meshes. This method divides the expression mesh into neutral

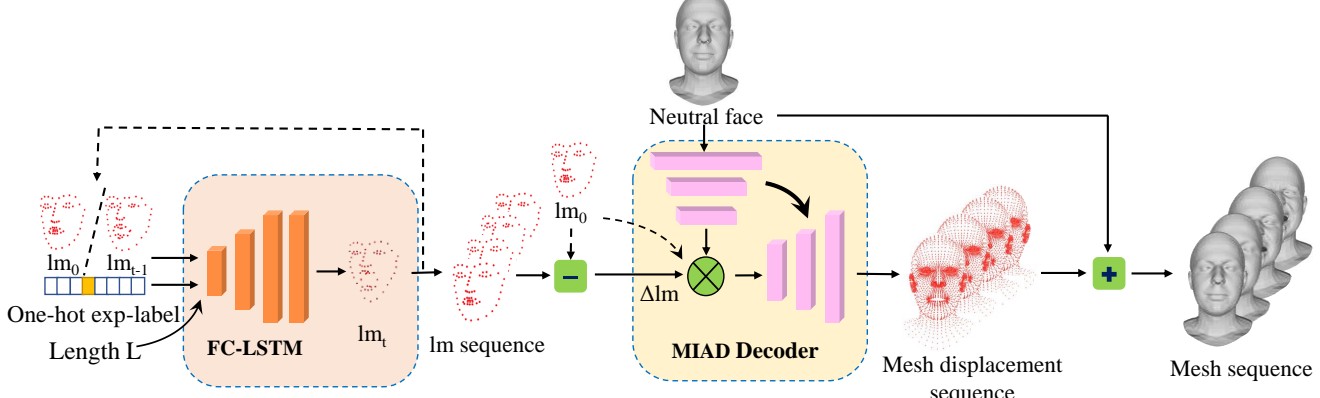

Figure 2: The overview of our FC-4DFS framework.

meshes and displacements and eliminates the need to encode and decode the main neutral mesh volume, greatly improving the ability to model expressions of unknown identities. However, due to the limited information contained in landmark displacements and the lack of diverse identity information, their ability to generalize the expressive mesh details of different identities still needs to be further enhanced.

## 3  METHOD

### 3.1  Overview

As shown in Figure.2, our FC-4DFS framework consists of a frequency-controlled LSTM that generates a landmark expression sequence and a MIADNet that transfers the landmark displacements into mesh vertex displacements. Specifically, the frequency-controlled LSTM uses the landmark $lm_{t-1}$ of frame $t-1$ and the expression label as input and generates the landmark $lm_t$ of frame t as output, where the neutral landmark $lm_0$ is given as the initial landmark at the first frame (See Section 3.2). Then, we subtract the initial neutral landmark $lm_0$ from the obtained landmark sequence $\{lm_t\}_{t=1}^n$ to obtain the displacement sequence for the subsequent network. Next, the MIADNET extracts multi-level latent features from the neutral mesh and predicts mesh displacement sequence with the input of landmark displacement sequence (See Section 3.3). Finally, we combine the mesh displacement sequence and the neutral mesh to obtain an expressional mesh sequence. Besides, in order to enhance the perception of temporal sequence motion and improve the accuracy of relative displacements, we also design a time consistency loss function (See Section 3.4).

### 3.2  Frequency-controlled LSTM

In this section, we propose a frequency-controlled LSTM, as shown in Figure. 3, to realize timing and frame rate perception of expression actions, realizing flexible frame-by-frame generation of variable-length sequences. Specifically, we first integrate frequency features into the LSTM for controlled long short-term memory updates, after which we introduce relative positional encoding embedding for free-framerate perception.

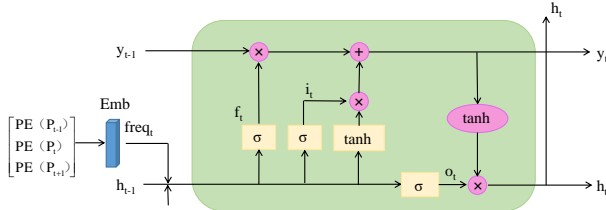

Figure 3: The structure of our frequency-controlled LSTM.

**Frequency Integration in LSTM** Given the input $x_t$ of the current frame $t$ extracted by the feature network and the output $y_{t-1}$ of the last frame $t-1$, the LSTM needs to obtain the output $y_t$ and update the hidden state $h_t$ in the model. In general, it first calculates the forget gate $f_t$ and the input gate $i_t$ as follows:

$$f_t = \sigma\left(W_f \cdot [h_{t-1}, x_t] + b_f\right)$$
$$i_t = \sigma\left(W_i \cdot [h_{t-1}, x_t] + b_i\right) \tag{1}$$

where $W$ and $b$ are the weight matrices and bias coefficients, $\sigma$ is the sigmoid function.

Through the hidden state and the controls of input gate and forget gate, the LSTM achieves stable and efficient timing-dependent modeling. However, LSTM implicitly sorts the input sequence by index, and the interval between two adjacent frame inputs is always 1 when entering a sequence of different lengths. This is appropriate for inputs that only consider precedence, such as the position of each word in a sentence. However, it cannot perceive the similarity difference between the two frames at different frame rates in the input with frame rate attributes, such as an image sequence in a video.

To do this, we integrate frequency information $freq_t$ into the LSTM to control the current frame forget gate and input gate as

follows:

$$f_t = \sigma\left(W_f \cdot [h_{t-1}, x_t, freq_t] + b_f\right)$$
$$i_t = \sigma\left(W_i \cdot [h_{t-1}, x_t, freq_t] + b_i\right) \quad (2)$$

After that, we obtain the candidate output $\tilde{y}_t$ and the output gate $o_t$ as:

$$\tilde{y}_t = \tanh\left(W_c \cdot [h_{t-1}, x_t] + b_c\right)$$
$$o_t = \sigma\left(W_o \cdot [h_{t-1}, x_t] + b_o\right) \quad (3)$$

Finally, we update the last hidden state from $h_{t-1}$ to $h_t$ and calculate the output $y_t$ as follows:

$$h_t = o_t * \tanh(y_t)$$
$$y_t = f_t * y_{t-1} + i_t * \tilde{y}_t \quad (4)$$

**Framerate-aware Positional Encoding** Take the expression label and initial neutral landmark as inputs, to generate a landmark sequence under free-framerate control, it is necessary to obtain the current position in the complete sequence and its time change relative to the previous frame. Thus, we employ relative positional encoding to code the entire sequence. In particular, we first calculate the relative position of the current frame in the full sequence as:

$$\tilde{p}_t = \frac{p_t}{L} \quad (5)$$

where $p_t$ is the time point of the current frame in the complete sequence, and $L$ is the duration of the complete sequence. Then, we encode the relative positional encoding of the current frame $PE$ as:

$$PE(p_t)^{2j} = \sin\left(\frac{\tilde{p}_t}{10000^{2j/d}}\right)$$
$$PE(p_t)^{2j+1} = \cos\left(\frac{\tilde{p}_t}{10000^{2j/d}}\right) \quad (6)$$

where $j$ represents the dimension within the encoding vector, and $d$ is the total number of dimensions in the model.

Finally, we embed the positional encoding of adjacent frames into the network as:

$$freq_t = Emb(PE(p_{t-1}), PE(p_t), PE(p_{t+1})) \quad (7)$$

which captures the temporal changes between the current frame and the previous frame, and realizes the free-framerate control for sequence generating.

## 3.3 MIADNet

The FC-LSTM previously generated realistic sequences of facial expression landmarks. However, our ultimate objective is to animate the neutral mesh $M_0$ to produce novel 3D facial animation sequences $\{M_t\}_{t=1}^n$. In this section, we propose a multi-level identity-aware displacement network (MIADNet) to make full use of the identity information from the generated landmark sequence and the neutral mesh, thus further enhancing the robustness to identity when reconstructing 4D facial expression sequence from landmark sequence. Specifically, the proposed MIADNet model, as shown in Figure. 4, includes a landmark decomposition embedding module, an identity extractor, and an identity-aware mesh generator.

**Landmark Decomposition Embedding** The generated landmark sequence after FC-LSTM contains identity and expression change information, where the identity information is shared throughout the action cycle, and provides a facial datum shape reference for the generation of the expressional mesh. We first decompose the

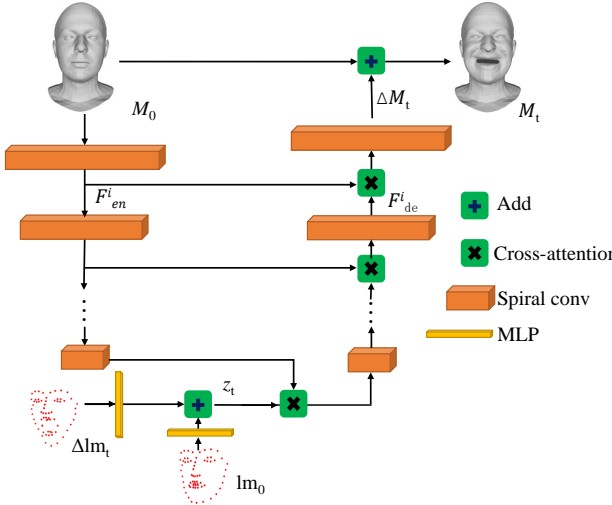

**Figure 4: The structure of our Multi-level Identity-Aware Displacement Net.**

generated landmark sequence $\{lm_t\}_{t=1}^n$ to the neutral landmark $lm_0$ and a landmark displacement sequence $\{\Delta lm_t\}_{t=1}^n$. For each frame $t$, we embed $lm_0$ and $\Delta lm_t$ to vectors in the latent code space with MLPs as the input to subsequent mesh generators, which is expressed as:

$$z_t = MLP_m(\Delta lm_t) + \lambda MLP_n(lm_0) \quad (8)$$

where $\lambda$ is a learnable parameter.

**Identity Extractor** The neutral landmark $lm_0$ provides global identity information at low resolutions, however, it doesn't have the ability to provide facial details for high-resolution expression animation meshes. In addition, it is difficult to maintain stability throughout the sequence for mesh generation using a single-frame landmark. For this purpose, we introduce neutral mesh $M_0$ as input, and we use multiple spiral convolutions to extract multi-resolution identity features from $M_0$. The multi-resolution feature constructs the intermediate bridge between low-resolution landmarks and high-resolution meshes, effectively improving the consistency of the generation process from landmarks to meshes and the stability in different frames.

**Identity-aware Mesh Generator** Given the latent landmark code $z_t$ and the multi-resolution identity features $F_{en}$ of the current frame $t$, the mesh generator upsamples $z_t$ and combines it with the identity features $F_{en}$ by skip connections to obtain the mesh output. To utilize the neutral mesh information as a reference mesh in the generation to reconstruct a continuous and consistent sequence, we use the cross-attention mechanism to model the context dependence between the generated expression mesh and the reference neutral mesh. Specifically, the input features of the i-level decoder can be expressed as:

$$F_{in}^i = \text{softmax}\left(\frac{F_{de}^{i-1} \cdot F_{en}^i}{\sqrt{d}}\right) F_{en}^i \quad (9)$$

where $F_{de}$ are the decoder features, and $d$ is the feature dimension.

## 3.4 Training loss

For the generation of facial expression animations, it is crucial to ensure the high quality of each frame and maintain continuity and smoothness throughout the sequence. This requires designing a loss function that takes into account the generation quality of each frame and the relationship between frames to solve these problems. Previous methods that directly generate complete sequences often directly measure the quality of the generated sequence to ensure accuracy and perceived fidelity. But this method tends to ignore motion in sequences. To this end, we propose a hybrid reconstruction and temporal coherence loss function to evaluate the generation quality of each frame and inter-frame motion respectively, while maintaining the generation quality while strengthening the model's modeling of sequence motion. The training loss of our model can be formulated as:

$$L_{total} = L_{re} + \alpha * L_{temporal} \tag{10}$$

where $L_{re}$ represents the single-frame reconstruction loss based on L1 distance, i.e.,

$$L_{re} = \frac{1}{N} \sum_{i=1}^{N} \|x_t - \tilde{x}_t\|_1 \tag{11}$$

and we set the hyperparameters $\alpha = 0.3$ in the loss function during training. Then, we use landmark motion between adjacent frames to enhance control over the smoothness of the sequence, and the temporal coherence loss $L_{temporal}$ is described as:

$$L_{temporal} = \frac{1}{N} \sum_{i=1}^{N} \|m_t - \tilde{m}_t\|_1 \tag{12}$$

## 4 EXPERIMENTS AND RESULTS

### 4.1 Experimental details

**Datasets** We carry out experiments on the CoMA[30] dataset and Florence4D [29] dataset. **CoMA dataset** comprises 144 facial expression sequences with 20,466 facial meshes collected from 12 unique subjects showcasing 12 different expressions, highlighting a vast spectrum of extreme expressions and the diversity in facial movements.

The CoMA dataset faces challenges due to data schema when applied to the task of generating label-driven facial expression animations. This is mainly because such tasks require a complete sequence from neutral expression to peak expression, and expression patterns(some are from neutral to peak while others are from neutral to peak and back to neutral) and the sequence lengths in the CoMA dataset are not uniform, and most of the frames of the sequence are concentrated in neutral and peak.

Existing methods, such as Motion3D, utilize sampling and interpolation to produce GT sequences, which struggle to smooth transitions and ignore motion changes of neutral and peak expressions. Therefore, we conduct a thorough refinement for the CoMA dataset to obtain the **Aligned CoMA dataset**. We first normalized sequence lengths, then emphasized significant expression changes, and curated the sequence to ensure the generation of animations that authentically and smoothly capture the nuances of facial expressions.

**Florence4D dataset** provides an extensive collection of 6,650 facial sequences with 399,000 face meshes from 95 distinct identities, encompassing 43 females and 52 males, and capturing a broad array of 70 unique facial expressions. Each sequence transitions from a neutral expression to an apex expression and then returns to neutral, demonstrating a rich diversity of facial movements and expressions. Both datasets are aligned in FLAME topology and have N = 5, 023 vertices.

**Implement Details** To maintain the independence of FC-LSTM and MIADNet, we train them separately on these datasets. For the landmark sequence generation of FC-LSTM, we first allocated the first three-quarters of the aligned datasets from CoMA and Florence4D for training, reserving the final quarter for testing. Meanwhile, we utilize the Adam optimizer and set the learning rate at 1e-5. Finally, we train 1,200 epochs for CoMA and 600 epochs for Florence4D. To rigorously evaluate the generalization capability of the MIADNet, we first implement a 4-fold cross-validation protocol for both the CoMA and Florence4D datasets, same as S2D[27]. Meanwhile, we utilize the Adam optimizer for training with a learning rate of 1e-3 and a decay rate of 0.99. Then, we train 300 epochs for the CoMA dataset and 150 epochs for the Florence4D dataset with a batch size of 128.

These experiments were meticulously designed to thoroughly examine the proposed framework's effectiveness and adaptability in capturing the complexities of facial expressions. Both tasks are trained on a device with 8 NVIDIA 4090 GPUs and Intel(R) Xeon(R) Platinum 8368 CPU @ 2.40GHz.

### 4.2 Comparisons

**Generating sequences of 4D facial expressions** In this section, we compare our FC-4DFS with the state-of-the-art methods including Motion3D[27] and LM-4DGAN[21]. Motion3D uses SRVF to encode landmark displacement and their open-source code is prone to crashing during the training stage. Therefore, we use their public model on the CoMA dataset for a 30-frame sequence comparison. Furthermore, we retrained the LM-4DGAN model on two datasets for comparison. Then, we generated the landmark sequence and reconstructed the mesh sequence from them, and used the mean square error (MSE) to compare the vertex reconstruction errors in landmark and mesh, $E_{lm}$(mm) and $E_{mesh}$(mm) respectively. Finally, to characterize the perceptual accuracy of the reconstructed expression sequences relative to expression labels, we trained expression-labeled classifiers on various datasets and used classification accuracy(CA) to test the generated result by various methods.

**Table 1: Quantitative results of the end-to-end comparison study.**

| Method | CoMA | | | Florence4D | | |
|---|---|---|---|---|---|---|
| | $E_{lm}$ | $E_{mesh}$ | CA(%)↑ | $E_{lm}$ | $E_{mesh}$ | CA(%)↑ |
| gt | - | - | **86.84** | - | - | **81.07** |
| Motion3D | 11.25 | 5.288 | 78.28 | - | - | - |
| LM-4DGAN | 10.02 | 4.724 | 75.30 | 9.349 | 8.414 | 69.37 |
| FC-LSTM+S2D | **8.308** | 4.392 | 83.79 | **7.496** | 6.642 | 76.45 |
| ours | **8.308** | **4.131** | **84.17** | **7.496** | **6.215** | **78.93** |

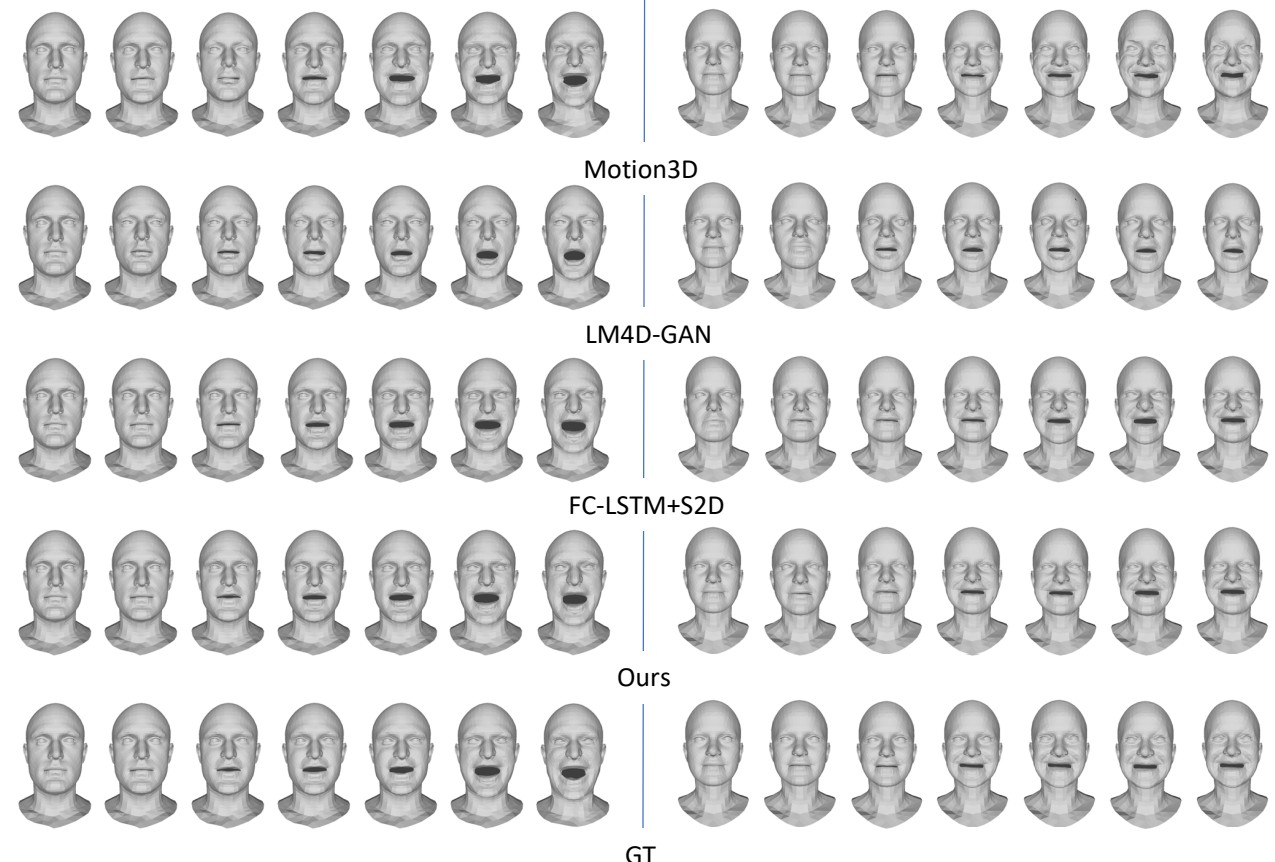

**Figure 5: The qualitative results of sequences generated by Motion3D, LM-4DGAN, our method and the ground-truth of different subjects.**

The quantitative comparison results are shown in Tab. 1. For 30 frame comparisons, the landmark per-vertex reconstruction error ($E_{lm}$) using our FC-LSTM is 26% higher than Motion3D and 17% higher than LM-4DGAN. At the same time, when using the same S2D decoder for end-to-end reconstruction, the mesh reconstruction error ($E_{mesh}$) per vertex is 18% higher than Motion3D and 7.6% higher than LM-4DGAN. In addition, when using our MIADNet to reconstruct the 4D facial expression from the generated landmark, it is improved by 21.8% compared to Motion3D and 12.5% compared to LM-4DGAN. The classification accuracy of the data generated by our method is presented in the third column of the table, demonstrating that the sequences produced, with each frame's mesh passing through a classification network, achieve results closer to the specified expressions. LM-4DGAN enhances the modeling of different identity movements due to the direct generation of landmark sequences, and its results are better than Motion3D. On the basis of utilizing neutral landmarks, our FC-LSTM generates sequences frame by frame and further considers the smoothness of inter-frame motion. Furthermore, our MIADNet introduces a neutral mesh with the cross-attention mechanism guided, which also improves the decoding result. Therefore, our method achieves the SOTA performance in all indicators and is good at generating label-driven 4D facial expression sequences with enhanced fidelity.

In addition, we qualitatively compared the generation results of our method with Motion3D and LM-4DGAN on multiple subjects at 30 frames. As illustrated in Figure5, the Motion3D sequence lacks continuity between frames, and detailed information about the mouth is missing in the sequence. LM-4DGAN uses a step-by-step generation method to enhance the continuity of the sequence and directly models the complete sequence from the landmark sequence, while it is more difficult to model motion patterns. In contrast, methods using FC-LSTM to generate the next frame based on the previous frame and expression labels, can achieve smooth and effective motion pattern modeling. Moreover, our final results with MIADNet achieve closer to GT than using the S2D decoder.

We mainly compared our MIADNet with several 3DMM-based methods, including the basic PCA-3DMM[2], the Neural 3DMM (N3DMM)[3] that specifically designed to learn the nonlinear latent space of mesh changes and reconstruct the input 3D mesh and the basic S2D[27] decoder without neutral identity information for the input displacement.

**Landmark Decoder** MIADNet is another important part of our FC-4DFS. In order to verify the robustness of the reconstruction

**Table 2: Quantitative results of the comparison study and ablation study for MIADNet.**

| Method | CoMA | Florence4D |
|---|---|---|
| PCA-220 | 1.280 ± 1.061 | 2.389 ± 1.660 |
| Neural3DMM | 3.873 ± 3.169 | 5.180 ± 6.635 |
| S2D | 0.550 ± 0.636 | 1.493 ± 1.607 |
| S2D+lm-n | 0.547 ± 0.637 | 1.405 ± 1.541 |
| S2D+mesh-n | 0.548 ± 0.635 | 1.421 ± 1.461 |
| Ours | **0.528 ± 0.634** | **1.338 ± 1.499** |

network to unseen identities, Refers to the experiment of S2D[27], we performed 4-fold cross-validation on the identities on the CoMA and Florence4D datasets and used per-vertex Euclidean error (mm) to measure the reconstruction performance. We compared our MI-ADNet with the basic S2D decoder and 3DMM-based methods, including the basic PCA-3DMM[2], the Neural 3DMM(N3DMM)[3].

The quantitative results of these comparisons are detailed in Table. 2. First, the spiral convolution-based N3DMM, which is proposed to learn the nonlinear latent space of facial changes for the 3DMM mesh with FLAME topology, performs poorly on both datasets. This is because N3DMM encodes and decodes the entire mesh, while it demonstrates some generalization capabilities across different expressions, its overall performance in handling new identities is still poor[27]. Furthermore, S2D decomposes the mesh into a neutral mesh that represents the identity and displacement, then uses the spiral convolution method to decode the displacement from the landmark into the displacement of the mesh. This method avoids encoding and decoding of neutral mesh to reduce errors but misses the use of identity information while its robustness to different identities needs to be further enhanced. Finally, our MIADNet adds the multi-level identity information for the S2D decoder, strengthening the reconstruction network's modeling capabilities for unknown identities. Therefore, our MIADNet achieve a 5% improvement on the CoMA dataset and 10.3% improvement on the Florence4D dataset compared to S2D. This shows that multi-level identity information can enhance the robustness of different identities.

On the other hand, we visualize the error heatmaps of the meshes reconstructed by these methods to compare their performance. Figure.6 provides qualitative examples of error heatmaps for identity-independent splitting by comparing them with PCA, N3DMM, and S2D. First, PCA-3DMM and N3DMM show high errors even for neutral meshes, demonstrating their inability to recover the identity of unseen meshes. Meanwhile, the high error of N3DMM is mainly concentrated in the lower half of the mesh, while the high error area of PCA is scattered throughout the mesh. This is because N3DMM takes advantage of the geometric topological relationship of 3DMM and uses spiral convolution to encode neutral meshes in the latent space, which can better reconstruct the topological structure of the mesh. Then, the S2D method and our method achieve lower reconstruction errors, which indicates that learning per-point displacements is better than directly learning point coordinates. Finally, our method shows significant improvement over the S2D method in handling highly varying expressions (see the neck and

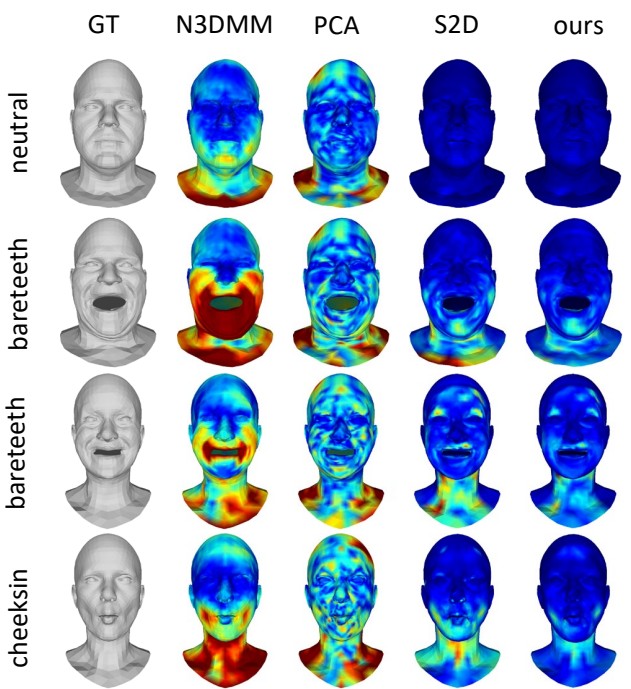

**Figure 6: Mesh reconstruction error (red=high, blue=low) of our model and other methods.**

eyes in Figure.6), which are consistent with the results shown in Table.2.

## 4.3 Ablation Study

**The Impact of Our Frequency-Controlled LSTM** To validate the role of LSTM in our proposed framework and the effectiveness of our novel frequency-controlled LSTM, we first use a basic MLP as the baseline for sequence regression and compare it with standard LSTM and our FC-LSTM. Due to the issues with the CoMA dataset mentioned in Section 4.1, to better compare the generated results, we trained on the aligned CoMA dataset(containing multiple sequence lengths) and the Florance4D dataset, and tested on 3 different lengths, as detailed in Table. 3. Furthermore, we use the first 3/4 identities as training data and the last 1/4 identities as test data and use per-vertex reconstruction error for evaluation.

It is observed that the framework utilizing LSTM for regression consistently achieves lower per-vertex reconstruction errors across all test lengths compared to MLP. Moreover, our FC-LSTM improves performance by 14% over the baseline in the aligned CoMA dataset and by 13.8% over the baseline in the Florence4D dataset. Compared with the basic LSTM, the improvement is 4.3% in the aligned CoMA dataset and 4.1% in the Florence4D dataset.

On the other hand, Figure.7 highlights the effectiveness of our FC-LSTM, demonstrating that our model can generate complete sequences of any given length and maintain fine detail across different lengths. Furthermore, the LSTM method without frequency

**Table 3: Ablation Study of FC-LSTM and temporal loss.**

| base | freq-info | loss-temp | Aligned CoMA | | | Florence4D | | |
|------|-----------|-----------|------|------|------|------|------|------|
| | | | 20 | 25 | 30 | 20 | 25 | 30 |
| mlp | | | 9.631 ± 4.919 | 9.533 ± 4.837 | 9.591 ± 4.815 | 8.796 ± 4.820 | 8.783 ± 4.727 | 8.790 ± 4.193 |
| lstm | | | 8.663 ± 3.641 | 8.577 ± 3.510 | 8.611 ± 3.580 | 7.920 ± 4.244 | 7.896 ± 4.244 | 7.894 ± 4.692 |
| lstm | | ✓ | 8.341 ± 3.676 | 8.222 ± 3.495 | 8.232 ± 3.500 | 7.686 ± 4.175 | 7.572 ±4.138 | 7.529 ± 4.124 |
| lstm | ✓ | | 8.291 ± 3.485 | 8.238 ± 3.485 | 8.255 ± 3.541 | 7.699 ± 4.167 | 7.603 ± 4.127 | 7.569 ± 4.102 |
| lstm | ✓ | ✓ | **8.152 ± 3.515** | **8.064 ± 3.446** | **8.058 ± 3.459** | **7.496 ± 4.082** | **7.411 ± 4.050** | **7.381 ± 4.031** |

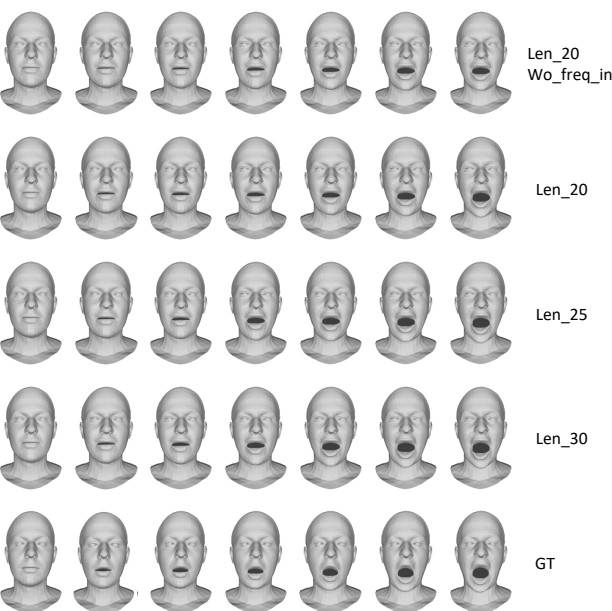

**Figure 7: Qualitative comparison results between sequences of various lengths generated using our FC-LSTM, sequences of 20 frames generated without frequency control, and the ground truth.**

control cannot effectively generate a complete expression sequence such as the mouth cannot be fully opened in the first row.

**Impact of temporal loss** Sequence continuity is of paramount importance for the task of 4D facial expression generation. The displacement between adjacent frames in a sequence represents the motion of landmarks throughout the sequence, and to smooth this motion, we incorporated a temporal loss into our training process. As shown in Table.3, the LSTM with added temporal loss exhibited a performance increase of 4.5% on the Align CoMA dataset. Furthermore, it has an improvement of 2.3% after adding the temporal loss for our FC-LSTM. On the Florence4D dataset, the implementation of temporal loss resulted in an enhancement of 4.6% for the LSTM framework and 2.5% for our FC-LSTM.

To further discuss the impact of hyperparameter $\alpha$, we compared the per-vertex reconstruction error at different frame rates when training with different $\alpha$ values. The experimental results are shown

in the Table.4 shows that the lowest reconstruction error is obtained when $\alpha$ is 0.3.

**Impact of MIAD** In the MIADNet part, based on S2D, we introduced Landmark Decomposition Embedding and Identity-aware Mesh Generator to extract multi-level identity information carried in neutral landmarks and neutral meshes respectively. To illustrate the role of multi-level identity information, we performed ablation experiments, and the results are shown in Table.2. It can be seen that adding Landmark Decomposition Embedding(S2D+lm-n) or Identity-aware Mesh Generator(S2D+mesh-n) alone does not significantly improve the reconstruction error. However, after introducing these two modules at the same time, we achieved a 5% improvement on the CoMA dataset and a 10.3% improvement on the Florence4D dataset compared to S2D.

**Table 4: Quantitative results of the ablation study for hyperparameters $\alpha$.**

| $\alpha$ | 20 | 25 | 30 |
|------|------|------|------|
| 0 | 8.291 ± 3.485 | 8.238 ± 3.485 | 8.255 ± 3.541 |
| 0.1 | 8.198 ± 3.700 | 8.125 ± 3.484 | 8.157 ± 3.547 |
| 0.2 | 8.282 ± 3.545 | 8.181 ± 3.489 | 8.173 ± 3.538 |
| 0.3 | **8.152 ± 3.515** | **8.064 ± 3.446** | **8.058 ± 3.459** |
| 0.5 | 8.276 ± 3.626 | 8.126 ± 3.444 | 8.081 ± 3.411 |
| 1 | 8.334 ± 3.596 | 8.240 ± 3.530 | 8.234 ± 3.561 |

## 5 CONCLUSION

In this paper, we propose a label-guided 4D facial expression synthesis method, FC-4DFS, by using frequency-controlled LSTM networks (called FC-LSTM) to form a flexible generation framework. This method not only overcomes the limitation that traditional methods can only generate fixed sequences but also further enhances the smoothness of generated sequence motion. The introduction of MIADNet further enhances our framework's ability to accurately reconstruct facial expression sequences with complex identity information from facial landmarks. Experiments conducted on the CoMA and Florence4D datasets demonstrate that our model achieves state-of-the-art performance.

It is worth noting that our method still needs to generate a landmark sequence first and then expand it to a mesh sequence according to the displacement. In future work, we will further study the end-to-end generation of 4D facial expression sequences.

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
