# OpenReview forum: "FC-4DFS: Frequency-controlled Flexible 4D Facial Expression Synthesizing"
_acmmm.org/ACMMM/2024/Conference — MM2024 Poster_

### Official Review · Reviewer_dyXd · 2024-05-23

**Rating:** 5
**Confidence:** 2

**Summary:**

The paper discusses a method called FC-4DFS, which is a frequency-controlled 4D facial expression synthesizing method. The method uses a frequency-controlled LSTM network to generate 4D facial expression sequences frame by frame from a given neutral landmark with a given length. It also introduces a temporal coherence loss to enhance the perception of temporal sequence motion and improve the accuracy of relative displacements. Additionally, the paper describes a Multi-level Identity-Aware Displacement Network based on a cross-attention mechanism to reconstruct the 4D facial expression sequences from landmark sequences. The proposed method outperforms SOTA methods on CoMA and Florence4D datasets.

**Strengths:**

1. The authors introduce a framework called FC-4DFS, which comprises a frequency-controlled LSTM (FC-LSTM) network to generate 4D facial expression sequences at frame-by-frame level and a Multi-level Identity-Aware Displacement Network (MIAD) based on a cross-attention mechanism to reconstruct the 4D facial expression sequences from landmark sequences.

2. The authors also perform extensive ablation studies and experiments on their proposed method using benchmark datasets and compare them with state-of-the-art methods.

3. In addition, the paper provides detailed explanations and descriptions of the proposed method and obtained results. Qualitative comparisons are also provided to substantiate FC-4DFS's performance.

**Limitations:**

1. The authors did not experiment with the D3DFACS dataset, which is used in the current state-of-the-art work S2D[27].

2. The authors should use the up-arrow ($\uparrow$) and down-arrow ($\downarrow$) to indicate the directions of all metrics in the table results. For example, for vertex reconstruction errors in landmark ($\downarrow$), lower is better.

**Suitability:**

3

---

### Official Review · Reviewer_Q6Dg · 2024-05-24

**Rating:** 4
**Confidence:** 4

**Summary:**

The authors propose a frequency-controlled LSTM network (FC-LSTM) to generate 4D facial expression sequences frame by frame and enhance their temporal coherence. A Multi-level Identity-Aware Displacement Network (MIADNet) is also designed to reconstruct the 4D facial expressions from landmark sequences. The method achieves state-of-the-art (SOTA) results on CoMA and Florence4D datasets, demonstrating its effectiveness in generating realistic and flexible 4D facial expression sequences.

**Strengths:**

Integrating FC-LSTM for frame-by-frame generation and MIADNet for identity-aware reconstruction is a creative solution that addresses the limitations of previous methods. Introducing frequency-controlled mechanisms and cross-attention in MIADNet shows a deep understanding of the technical challenges in 4D facial expression synthesis. Using CoMA and Florence4D datasets allows for a thorough evaluation of the method's capabilities.

**Limitations:**

The framework may be complex to implement, which could limit its adoption by other researchers and practitioners. While the method shows strong results on the tested datasets, it needs to be clarified how well it generalizes to other datasets or variations in facial expressions. The paper could benefit from a discussion on the computational efficiency and scalability of the proposed method. More detailed comparative analysis with other state-of-the-art methods would be beneficial, especially regarding computational resources and scalability.

**Suitability:**

3

---

### Official Review · Reviewer_P2Mw · 2024-05-24

**Rating:** 4
**Confidence:** 3

**Summary:**

The task aims to generate a series of realistic facial mesh with diverse expressions or speech-related movements, starting from a mesh with a neutral expression. In this domain, this paper describes a frequency-controlled 4D facial expression synthesizing method based on frequency-controlled LSTM and multi-level identity-aware network. This is combined with a multi-level identity-aware network based on a cross-attention mechanism to reconstruct identity-consistent 4D facial expression sequences from landmark sequences.

**Strengths:**

Well written, detailed analysis of previous related work. A novel aspect of the proposed investigation is the use of expression labels to guide a temporally coherent animation process, enhancing the applicability of facial expression sequence generation in scenarios with minimal priors.

**Limitations:**

The processing pipeline is composed of two cascaded modules: 1) generation of the landmark expression sequence and 2) transfer landmark sequence displacements to mesh vertex displacements through the MIADNet. It is not clear how this architecture is able to process a continuous input video stream: is it necessary to decompose the stream into blocks and process the first one with the first module before passing it to the second module? This may introduce some delay and compromize smooth animated transition across different blocks.
It is really hard to find a significant difference in the data reconstructed by differente techniques as reported in Fig.5

**Suitability:**

2

---

### Meta-Review · Area_Chair_t1Go · 2024-06-23

**Recommendation:** Accept (Poster)
**Confidence:** 5

**Metareview:**

The reviewers all felt positively about this paper. The paper is well written and detailed analysis of previous related work.